# Genetic Constructs for the Control of Astrocytes’ Activity

**DOI:** 10.3390/cells10071600

**Published:** 2021-06-25

**Authors:** Anastasia A. Borodinova, Pavel M. Balaban, Ilya B. Bezprozvanny, Alla B. Salmina, Olga L. Vlasova

**Affiliations:** 1Laboratory of Cellular Neurobiology of Learning, Institute of Higher Nervous Activity and Neurophysiology, Russian Academy of Sciences, 117485 Moscow, Russia; borodinova.msu@mail.ru; 2Laboratory of Molecular Neurodegeneration, Peter the Great St. Petersburg Polytechnic University, 195251 St. Petersburg, Russia; Ilya.Bezprozvanny@UTSouthwestern.edu (I.B.B.); allasalmina@mail.ru (A.B.S.); olvlasova@yandex.ru (O.L.V.); 3Department of Physiology, University of Texas Southwestern Medical Center at Dallas, Dallas, TX 75390, USA; 4Research Institute of Molecular Medicine and Pathobiochemistry, V.F. Voino-Yasenetsky Krasnoyarsk State Medical University, 660022 Krasnoyarsk, Russia; 5Research Center of Neurology, 125367 Moscow, Russia

**Keywords:** astrocytes, AAV, opsins, viral vector, serotype, glia, promoter, GFAP

## Abstract

In the current review, we aim to discuss the principles and the perspectives of using the genetic constructs based on AAV vectors to regulate astrocytes’ activity. Practical applications of optogenetic approaches utilizing different genetically encoded opsins to control astroglia activity were evaluated. The diversity of astrocytic cell-types complicates the rational design of an ideal viral vector for particular experimental goals. Therefore, efficient and sufficient targeting of astrocytes is a multiparametric process that requires a combination of specific AAV serotypes naturally predisposed to transduce astroglia with astrocyte-specific promoters in the AAV cassette. Inadequate combinations may result in off-target neuronal transduction to different degrees. Potentially, these constraints may be bypassed with the latest strategies of generating novel synthetic AAV serotypes with specified properties by rational engineering of AAV capsids or using directed evolution approach by searching within a more specific promoter or its replacement with the unique enhancer sequences characterized using modern molecular techniques (ChIP-seq, scATAC-seq, snATAC-seq) to drive the selective transgene expression in the target population of cells or desired brain regions. Realizing these strategies to restrict expression and to efficiently target astrocytic populations in specific brain regions or across the brain has great potential to enable future studies.

## 1. Introduction

It is a common opinion now that animal behavior results from the coordinated activity of neurons and astrocytes that are actively involved in synaptic physiology, regulation of long-term changes in brain functioning including neuropathologies. Having in mind the idea of compensation of pathological changes in the neural networks via activation of glia, new approaches are constantly suggested. An effective approach to achieving selective glia stimulation in vivo is to deliver genetically encodable chemogenetic or optogenetic effectors whose expression is restricted by glia-specific promoters using recombinant AAV vectors. Targeted delivery of the relevant ligand for chemogenetic effectors or focused illumination for opsins can further restrict glia stimulation to the desired population. In the present paper, we discuss the results and the problems essential for the effective stimulation of glial cells in vivo, where this glial stimulation is designed to influence neuronal function.

## 2. Astroglial Molecular Physiology

### 2.1. Heterogeneity of Astroglia

Astroglia play numerous essential roles in the brain. Decades of research demonstrated that astrocytes regulate neuronal excitability and metabolism, establish and regulate the blood–brain barrier, locally control microcirculation, contribute to neuroinflammation, regulate neurogenesis, support brain tissue clearance, maintain water homeostasis, mediate the exchange between cerebrospinal fluid and interstitial fluid in the brain tissue, and control edema formation in an AQP4-dependent manner [1,2,3,4,5,6]. The aberrant functioning of astroglia was observed in neurodevelopmental and neurodegenerative diseases, brain injury, and neuroinflammation. Given these observations, the modulation of astrocytes affects the progression of brain diseases and is considered a promising neuroprotective and neuroregenerative therapy [4,7,8,9]. However, targeted modulation of astroglial activity faces numerous problems. First, various subpopulations of astrocytes are involved in the functions mentioned above, making it difficult to specifically target the relevant astroglia population. Second, astrocytes are very sensitive to neuronal activity and vice versa. This interdependence means that the activation of neurons and astrocytes influences each other, making it challenging to characterize and observe an independent glial response.

Astrocytes are a heterogenous group of brain cells with diverse molecular markers [10]. This diversity of molecular markers makes it challenging to identify astrocytes by detecting a single marker such as glial fibrillary acidic protein (GFAP) or s100β protein expression alone. There are several subgroups of astrocytes in rodent and primate central nervous systems, such as:
(i)radial astroglia (GLAST+(–)/GFAP(+)(–)/Pax6+/FABP7+/Nestin+/Vimentin+/Pax6+/Cx43+/Cx30 cells) originating from neuroepithelial cells that are involved in embryonic and adult neurogenesis;(ii)protoplasmic astrocytes (GFAP++/s100β+/EAAT+/AQP4+/NDRG2+/Cx43/CD38+), which are the principal glial constituents of the neurovascular unit; they stay close to neurons due to direct contacts made by their perisynaptic processes and control neuronal excitability, plasticity, metabolic status, and close to brain microvessel endothelial cells due to their end-feet contacts to adjust local microcirculation to the actual needs of neurons;(iii)fibrous astrocytes (GFAP+(–)/CD44+ cells) surrounding myelinated fibers and controlling myelinization; iv) reactive astrocytes with upregulated expression of GFAP (GFAP++/EAAT+/Nestin+/Vimentin+/PDGFR+/Musashi+/CD44+/CD38+/Lcn-2+ cells) that take part in the progression of local inflammation and gliosis [11,12,13,14,15].

Reactive astrocytes are classified into two types, A1 and A2. These two types are characterized by the predominant expression of pro-inflammatory (e.g., lipocalin 2, Lcn-2) or anti-inflammatory (e.g., thrombospondins, neuroprotective cytokines) molecules, respectively [16]. In addition, neural circuit-associated astrocytes might be a separate class, as suggested by proteomic, transcriptomic, and cell marker expression analysis [17].

Specific types of astrocytes were observed in the higher-order primate and human cortex: intralaminar astrocytes (with tangential and long vertical processes) and varicose projection astrocytes (with long fibers and varicosities) that are in close contact with brain capillaries that are presumably involved in long-distance intra-cortical communication [18]. Human and higher-order primate protoplasmic astrocytes are larger and express more GFAP than rodent protoplasmic astrocytes [19]. It is known that astrocytes have polarized patterns of expression either for GFAP or AQP4 (channel, regulating water exchange); in the latter case, anchoring of AQP4 in end-feet of perivascular astroglia is needed for the appropriate channel activity [20], however, most of the data were obtained either from 2D models cell culture or from animal studies. Thus, application of up-to-date protocols for the establishment of 3D culture, iPSCs-derived spheroids and organoids, humanized self-assembled cell models, and cell imaging is required for real-time monitoring of astroglial activity within the neurovascular unit and the blood–brain barrier [21,22,23]. It should be taken into consideration that the expression pattern of all these subgroups of astrocytes and their functional activity greatly depend on the stage of development and the brain region [24].

### 2.2. Measuring Astroglial Activity

Several approaches were developed to record and monitor astroglial activity. They are mainly based on the following phenomena:
(i)rise in intracellular Ca^2+^ concentrations due to Ca^2+^ release from intracellular stores, Ca^2+^ entry, or connexon-mediated propagation of Ca^2+^ waves [25];(ii)metabolic changes in astrocytes (lactate production, K^+^ uptake, mitochondrial activity, etc.) caused by their activation [26];(iii)changes in the expression pattern in activated glial cells, i.e., due to conversion of resting astrocytes into reactive ones [27];(iv)changes in astroglial morphology [28];(v)and changes in astroglial secretory activity resulting in the release of gliotransmitters, cytokines, and growth factors [29].

Release of lactate from astrocytes results in the metabolic support of activated neurons due to rapid conversion into pyruvate and fueling the mitochondria respiration (astrocyte-neuron lactate shuttle) [30,31] as well as in the modulation of blood–brain barrier integrity or initiation of angiogenesis/barriergenesis [32]. Even though there are some controversial data on the lactate-driven communication of astrocytes and neuronal cells [33], this mechanism is considered as a target for manipulating the astroglial activity in vitro or in vivo [7].

Numerous studies confirmed the release of gliotransmitters from activated astrocytes in physiological and pathological conditions that depends on different mechanisms. Ca^2+^ release-activated calcium channels (CRAC) and IP_3_R-regulated Ca^2+^ release from intracellular stores are required for vesicle exocytosis and ATP secretion from astrocytes [34]; the complex interplay between calcium stores and channels coordinates Ca^2+^-dependent exocytosis of glutamate-containing vesicles in astrocytes [35], and arachidonic acid-activated TRPV4 channels contribute to Ca^2+^ rise and release of ATP from astrocytes [36]. However, some recent data suggest that calcium dependence in the release of gliotransmitters in astrocytes associated with specific neural circuits might be overestimated: calcium signals are rather variable in their dependence on Ca^2+^ entry or Ca^2+^ release from intracellular stores in astrocytes in various brain regions, and the role of calcium in astroglial glutamate exocytosis is quite different in young and aging brains [17]. Moreover, some authors propose that the application of widely used methods to induce astroglial activation (for instance, strong depolarization in patch-clamp experiments, optogenetic stimulation, or agonist-induced activation of GPCRs) might not be reliable tools to reconstitute the physiological activity of astrocytes. Therefore, Ca^2+^-mediated release of transmitters seems to be exclusively neuronal in the healthy brain [37].

Given that Ca^2+^-mediated transmitter release is exclusively neuronal under physiological conditions, effective and physiological astroglial activation is a rather non-trivial task in experimental neurobiology. The experimental challenges are increased by the demonstration that the expression of activity-dependent genes (analog to the neuronal immediate early genes, IEGs) is evident in stimulated astrocytes [38,39]. This expression of activity-dependent genes might affect data obtained in the brain tissue or in the whole brain when IEGs were used as markers of neuronal activation [40]. In particular, a new population of astrocytes called immediate-early astrocytes (as a transition state between quiescent and reactive astroglia) with marked c-fos expression was recently identified in the brain perivascular areas in the animal model of multiple sclerosis [41]. Glutamate-induced expression of c-fos mediated by the mobilization of Ca^2+^ from the endoplasmic reticulum was reported in primary cortical rat astrocytes in vitro [38]; therefore, this phenomenon seems essential for cell-to-cell communication in the neurovascular unit. In addition to c-fos, another immediate-early gene Arc/Arg3.1 (activity-regulated cytoskeleton-associated protein) is overexpressed in activated astrocytes and demonstrates co-localization with GFAP in cell bodies, large- and medium-sized astroglial processes upon induction of LTP in neighboring neurons [39]. A similar phenomenon associated with a reduced Ca^2+^ influx could be achieved by exposure of astrocytes to extracellular lactate in physiological concentrations acting via GPR81 receptors through the β-arrestin2/MAPK-pathway [42]. It should be noted here that lactate produces upregulation of Arc/Arg3.1 expression in hippocampal neurons during learning [43], supporting the idea of the importance of lactate-generating astrocytes for the coordinated activation of neuronal and glial cells. This mechanism remains to be evaluated because some recent data suggest that Arc may not be transcribed in astrocytes but is transferred from the activated neuronal cells and then is closely associated with GFAP in astroglia [44].

In summary, a broad spectrum of astrocytic phenotypes should be considered when designing approaches to modulate targeted astroglial populations in vitro and in vivo. Specifically, experimentalists should select an appropriate genetic construct when using optogenetics to precisely target astroglial cells or when planning the expression of a reporter gene.

### 2.3. Choice of Genetic Constructs Suitable for Glia Activation

Neuronal activity elevates intracellular calcium concentration in nearby located glial cells following activation of G protein-coupled receptors by synaptically released neurotransmitters. Many approaches are used to investigate inverse changes in neuronal activity due to calcium waves in astrocytes and modulation of synaptic transmission through the release of gliotransmitters. Theoretically, selective regulation of astrocytes’ functioning with the aid of genetically encoded sensors can be made using chemogenetics, thermogenetics, and optogenetics. It was shown that activation of astrocytes using either a chemogenetic or an optogenetic genetically encoded tool during learning resulted in memory recall enhancement on the following day, suggesting the involvement of glial cells in plasticity regulation [45]. Most contemporary approaches are optogenetics based on a range of tools involving expression in astrocytes of specific light-sensitive opsins [46] or chemogenetic approaches using artificial receptors in astroglial cells [7].

There exist two kinds of opsins. Type I (microbial) opsins are ion channels or proton/ion pumps directly activated by light [47]. Type II (animal opsins) are members of the G protein-coupled receptors (GPCR) superfamily and activate G-proteins, leading to the activation of effector enzymes and the modulation of specific ion channels, resulting in mobilization of Ca^2+^ from intracellular stores [48,49]. Molecular mechanisms of glial activation by Type I and II opsins are different, and effects on neuronal activity may also differ.

As a first suggested tool for minimally invasive, genetically targeted, and temporally precise photostimulation, the algal protein channelrhodopsin-2 (ChR2) was a good candidate because of its temporal precision and spatial targeting [50]. ChR2 and its modifications, including hyperpolarizing and dual-action channelrhodopsins [51], are still the most widely used genetic constructs to control excitable cells and glia. Recently, a versatile family of genetically encoded opsins (“optoXRs”) that activate via G-proteins a range of biochemical signaling pathways in response to light was developed [52]. OptoXRs can have inhibitory or excitatory effects on neural activity and were shown to effectively stimulate Ca^2+^ increase in astrocytes. Expression of the Gq-coupled receptor hM3Dq in CA1 astrocytes, allowing their activation by a designer drug (chemogenetics), showed that astrocytic activation efficiently regulated the synaptic plasticity [45]. Different gliotransmitters such as glutamate, ATP, GABA, or D-serine released from astrocytes were shown to regulate various synaptic events and even long-term plasticity. Recently, it was shown that mouse hippocampal astrocytes activated by endogenous or exogenous (single astrocyte Ca^2+^ uncaging) stimuli modulate the efficiency of CA3-CA1 hippocampal synapses [53]. The initial effect (weak stimulation of astrocytes) was associated with the release of glutamate and potentiation of synaptic inputs, followed by a purinergic-mediated decrease of efficiency of CA1 synaptic inputs. The properties of this biphasic synaptic regulation depend on parameters of neuronal activity that affect astrocytes [53]. Data obtained in this study explain multiple opposing published results of potentiation or depression of synaptic effectivity under glia stimulation because parameters of astroglial stimulation might be significantly different. A promising optogenetic approach is based on selective expression of mammalian melanopsin, a G-protein-coupled natural photopigment, in astrocytes to trigger Ca^2+^ signaling. Recently, it was shown that melanopsin can be selectively expressed in astrocytes and serve as a precise trigger for glial cell activation that mimics their endogenous GPCR-driven signaling pathways. By releasing ATP, melanopsin-expressing astrocytes affect synaptic plasticity and can enhance cognitive functions in vivo [54].

Leaving aside the controversial reasoning about the mechanisms of gliotransmission shown above, we may assume that astrocyte stimulation in active brain regions where neurons demonstrate either excitatory (glutamate) or inhibitory (GABA) activity results in an intracellular increase in Ca^2+^ concentrations followed by glial activation to support or to inhibit neuronal activity. Considering that glia are integrators of a sophisticated set of transmitter receptors that lead to distinct secretion mechanisms, we should accept very slow timing of glia reactions if compared with neurons and slow effects mostly by diffusion with extremely high locality of astroglial effects [55]. It is essential to consider astrocytes as active information integrators and processors, reacting in each case according to the input signals with particular timing and location.

Recognition of astrocytes as integrators of inputs makes easier the task of external stimulation that can be performed via any receptors but stresses the importance of timing and locality of optogenetic or chemogenetic stimulation. It was shown that selective stimulation of astrocytes expressing channelrhodopsin-2 (ChR2) in the CA1 area elicits ATP release locally and specifically increases the firing frequency of CCK-positive but not parvalbumin-positive interneurons and decreases the firing rate of pyramidal neurons [56]. The induction of long-term potentiation (LTP) in tetanized inputs of neurons elicits a stimulation-induced ATP release from astrocytes and a local modulation of the untetanized synapses [57]. Simultaneously, the depression of synaptic inputs is elicited via activation of P2Y receptors, maintaining an excitation/inhibition balance in all inputs of the neuron, thereby preventing the “run-away” effect [57]. The mechanism of heterosynaptic interactions and the heterogeneity of synaptic strengths of pyramidal cell inputs in single hippocampal neurons were investigated, and the necessity of astrocytes for counterbalancing the converging synaptic input strengths was shown [58].

The success of the practical application of optogenetic approaches is directly related to the creation of correct vector constructs based on adeno-associated viruses (AAV) for the delivery of genetic material into glial cells. Employment of specific AAVs for directed cell-specific gene expression may be a challenging problem. Analysis of published data suggests that designing effective AAVs with specific properties depends on a combination of parameters that the investigator can assign to the construct. In the following sections, we discuss the obstacles and the perspectives of the usage of different AAV serotypes and specific promoters in the AAV cassette for directed glia-specific delivery of the genetic constructs.

## 3. Targeted Delivery of Genetic Constructs into Glial Cells

### 3.1. Is the Optimal AAV Serotype Myth or Reality?

The discovery of adeno-associated viruses (AAVs) in the late 1960s and the production of the first recombinant viruses in the late 1980s provided a basis for developing a powerful tool for gene delivery into the central nervous system [59]. Early studies provided comparative analyses of AAVs with known natural serotypes to describe how given recombinant viruses may influence the transduction efficacy of defined cell populations in particular brain regions [60,61,62,63].

Usage of the ubiquitous promoters (CAG, CMV, CBA, etc.) allowed analysis of the original properties (tropism) of many discovered AAV serotypes that were based on their capsid protein structure. Aschauer and colleagues provided in-depth in vivo screening of a set of recombinant AAVs (AAV1, 2, 5, 6, 8, 9) for four cell types (neurons, microglia, astrocytes, oligodendrocytes) in three different regions of the mouse brain [61]. They found that tested AAV viral vectors with a reporter gene driven from a ubiquitous promoter led to expression in diverse cell types in the brain. However, some viruses produced marked differences in the reporter gene expression levels in particular cell populations that correlated to specific serotypes. According to the literature, the most confident results were found for the AAV2 serotype that showed preferential neuronal tropism even when used in combination with ubiquitous (CMV, CBA, CBh) or glial promoters (hGFAP) [61,62,63,64,65]. Less consistent experimental data were found for glial preferences of several AAVs. Two independent studies suggested that either AAV5 or AAV8 was the most efficient serotype for astrocytic infection in the mouse hippocampus, having tested the similar naturally occurring serotypes, AAV1, 2, 5, 8, 9, and either 6 or 7 [60,61]. Another screening of a similar panel of recombinant AAVs in the rat cortex in vitro (AAV1, 2, 5–9) and in vivo (AAV2, 6) demonstrated the highest AAV6-mediated transduction of rat astrocytes [63]. Interspecies comparison of AAV5, 8, and 9 but not AAV1 or 2 in the cortex of non-human primates and mice revealed that the infected cells exhibited size and morphology characteristics of glial cells in both evolutionary groups [62]. However, the AAVs infected astrocytes more in mice and infected oligodendrocytes more in primates. In contrast, neuronal or neuronal/oligodendrocytic targeting was observed for AAV8 and AAV9 in rat striatum when various ubiquitous promoters were used [65,66]. AAVs capsids 1, 2, 4–6, 8, 9 tested in vitro on primary human astrocytes showed unexpectedly great cell-binding ability and the most efficient viral transduction for AAV2, while AAV8 was the least efficient [67]. To summarize, the AAVs with a reporter gene driven from diverse ubiquitous promoters differed in their astrocytic transduction efficacy that varied across brain regions and between the species.

Where do the discrepancies of observed AAV-mediated glial infection arise? Partially, the controversial data may arise from the naturally occurring differences of cellular characteristics and molecular machinery in astrocytic populations that may be critical for the AAV life cycle (endocytosis, transport to the nucleus, unpackaging of the viral DNA from the capsid, etc.) and efficient expression of a genetic construct. It should be noted that methodological differences may provide an additional source of the controversial data for AAV tropism. Several studies suggest that AAV-mediated tropism depends on the ontogenetic timing, and the efficacy of glial transduction by different AAV serotypes in vitro and in vivo may change with age [68,69,70]. Different ontogenetic timing of systemic injection (intravenous, intracerebroventricular) of AAVs in mice resulted in strikingly opposite transduction patterns observed in the brain regions: transient neuronal or neuronal/astrocytic targeting throughout the neonatal brain and predominantly astrocytic labeling throughout the brain at later stages of ontogenesis [69,70]. Potentially, such distinctive tropism of AAVs in the developing brain reflects an early (neurons) and a late (glia) appearance of target cell populations bearing the appropriate receptors for AAV capsids. Among the tested viruses (AAV1, 2, 5, 7–9), the most prominent shift in cellular tropism with age was found for AAV8 and AAV9. It is noteworthy that AAV5 demonstrated consistent widespread astrocytic transduction irrespective of the time of virus application in vivo and in vitro [68,70].

Another methodological issue relates to the AAV delivery approaches. The majority of published studies operated with local injections of viruses into particular brain areas. However, recent experiments tested minimally invasive systemic injections of suitable AAV serotypes to deliver genetic constructs to the brain [69,71,72,73,74]. Unexpectedly, the cellular tropism of particular AAV serotypes can be injection route-dependent, making it impossible to combine the data obtained from different protocols of local and systemic virus delivery [69,71]. Gray and colleagues observed AAV8-mediated transduction of neurons and oligodendrocytes after intravenous virus delivery [71], which contradicts the previously discussed astrocytic tropism of locally injected AAV8 (see above). Another group showed that intravenous injection of AAV9 serotype resulted in preferential astrocytic transduction, whereas locally injected AAV9 viruses exhibited a predominantly neuronal transduction pattern [69]. However, additional studies are required because the original data were not fully described in the paper mentioned above. Some differences in delivery were also observed for synthetic AAV serotypes (PHP.B, PHP.eB) generated based on AAV9.

Efficacy of AAV-based delivery depends on the specific receptors expressed in brain microvascular endothelial cells of the blood–brain barrier, mainly, lymphocyte antigen-6A (LY6A) receptor (which is absent in BALB/cJ mice) mediates binding and transfer of AAV-PHP.B capsids independently on AAV9 receptors [75]. Brain-targeted transduction with AAV-PHP.eB capsid possessing different amino acids at positions 587–589 of the AAV-PHP.B capsid sequence depends on the expression of low-density lipoprotein receptor (LDLR) in endothelial cells and shows better transduction efficacy in C57BL/6 compared with BALB/c mice [76]. These data suggest that mouse strain specificity might affect the transduction of brain cells with AAV-based vectors, but the aberrant expression of corresponding receptors in other brain cells should be considered. For instance, LY6A is also known as a stem cell antigen-1 (Sca-1) and belongs to the superfamily of lymphocyte antigen-6 (Ly6)/urokinase-type plasminogen activator receptor (uPAR) proteins [77]. Sca-1-positive vascular stem cells might have a role in vascular remodeling in the brain [78], suggesting altered transduction of AAV vectors in pathologies associated with deregulated angiogenesis. In Alzheimer’s disease, amyloid-β causes a reduction in the levels of released LDLR ligand ApoE and increased cell-associated ApoE expression in astrocytes in the LDLR-dependent manner [79]. This increased expression of LDLR could result in enhanced AAV transduction of astroglial cells in Alzheimer’s type neurodegeneration. However, these proposals require further investigations.

Another critical determinant of AAV-mediated cellular tropism may refer to the virus purity. The use of alternative protocols for AAV purification (CsCl or iodixanol gradients) resulted in significantly different transduction patterns observed for particular serotypes in vitro and in vivo [63,80]. According to Klein and colleagues, CsCl-purified AAV8 viruses displayed reproducible astroglial targeting compared to the similarly prepared AAV9 [80]. The change of purification method to iodixanol shifted cellular tropism of AAV8 viruses from astrocytic to neuronal. Subsequent protein analysis of AAV8 batches revealed the excessive virus impurities in the CsCl-purified virus suspension that potentially alter cellular tropism. Schober and colleagues found inconsistent AAV-mediated astrocytic tropism between viral batches from different packaging facilities and even different viral batches from the same packaging facility [63]. It might be the case that variability in virus preparation and purification protocols produces different protein impurity levels that interfere with proper capsid–receptor interactions on the cell surface at different degrees, leading to less specific virus targeting.

To summarize, the astroglial tropism of AAV viruses and transduction efficacy can vary across serotypes with exceptional variability for AAV8 and AAV9, depending on experimental conditions of timing and route of virus injection, AAV production protocols, and region- and cell-specific characteristics in the injection area. Therefore, AAV-mediated expression of a genetic construct in particular cell types requires an additional strengthening that involves cell-specific promoters and diverse regulatory elements in the AAV cassette.

### 3.2. General Requirements for the AAV Cassette for Efficient Transgene Expression

The genome of wild-type and recombinant AAVs consists of linear single-stranded DNA flanked by palindromic inverted terminal repeats (ITRs), which are necessary for packaging the DNA fragment into the viral capsid. The combination of AAV2-based ITRs (present in most of the available AAV plasmids) with appropriate capsid proteins is generally used to produce various recombinant AAV pseudotypes for neurobiological studies (see above). Compared to other viruses, recombinant AAVs are characterized by a relatively small packaging capacity of about 4.7 kb for sure (including two ITRs, 145 bp each) that corresponds to or insignificantly exceeds the length of the wild-type AAV genome. Considerable excess of the recommended genome length may affect the production of functional AAV virions, e.g., infectious titer [81].

Considering the substantial size of diverse glial-specific promoters and regulatory elements (see below), it is challenging to overexpress large genes (or genes with fluorescent reporters) in selected populations of cells without compromising AAV infectivity, specificity, or transcription efficacy. Typically, the length of tested astrocytic promoters with relatively high cell specificity is around 0.7–2.6 kb (Table 1; see below), which leaves around 1.8–3.7 kb vacant. However, there were some precedents where AAVs can accommodate the insertions with larger or smaller sizes depending on the promoter chosen [64]. It was also noted that specific, sequence-dependent capsid–promoter interactions might take place, interfering with resulting AAV properties such as cell-specific gene expression profile [65].

Previously, it was shown that specific promoters exhibit different activities ranging from strong to weak [64,82]. Incorporating diverse post-transcriptional regulatory elements and polyadenylation signals (polyA) in the 3’-flanking region of the target gene can also enhance the efficacy and the duration of its expression through the regulation of mRNA nuclear export, stability, and translation [83,84,85,86,87,88]. It was found that insertion of the most effective Woodchuck hepatitis virus Post-transcriptional Regulatory Element (WPRE, 0.6 kb, Table 1) into the AAV cassette considerably increased transgene expression in neuron cultures and brain regions of rodents (1.8- to 35-fold depending on the experimental conditions) [64,89,90]. Choi and colleagues demonstrated that the presence of efficient polyadenylation signals bGH or SV40 late polyA (Table 1; 223 and 135 bp, respectively) in the AAV cassette can additionally elevate to a similar degree the transgene expression in mouse neurons [83,84,90]. Therefore, utilization of standard regulatory elements for improved transgene expression allows incorporating the fragment of about 1–2.9 kb into the AAV cassette. Visualizing transgene expression with diverse fluorescent reporter proteins (EGFP, EYFP, mCherry, etc., Table 1) occupies an additional 0.7 kb, leaving only 0.3–2.2 kb for the target gene. The insertion of larger fluorescent proteins, such as 1.4 kb tdTomato, can exceed the packaging capacity of AAV constructs driven by long promoters.

Genetic engineering of WPRE and late SV40 regulatory elements presented recently by Choi and colleagues resulted in the development of effective equivalents with a smaller size (Table 1) that provided an additional 0.4 kb extra space in the AAV cassette [90]. Therefore, ongoing rational optimization of the AAV cassette with compact and strong astrocytic promoters or post-transcriptional regulatory elements has undeniable potential.

### 3.3. GFAP as a Specific Marker of Astroglial Cells

The most popular astrocyte marker, regardless of the astrocyte cell-type of interest, is GFAP (glial fibrillary acidic protein), a cytoskeletal intermediate filament protein expressed in mature astrocytes and radial glia. GFAP mutations result in protein deposits known as Rosenthal fibers in Alexander disease [91]. The GFAP gene is organized as nine exons and, in the human genome, is approximately 10 kb in length. Alternative mRNA splicing generates different GFAP isoforms [92]. GFAP isoforms have several functions, including signal transduction and integration in astroglial cells and the stabilization of astroglial processes through the interaction of GFAP with vimentin [93,94,95]. Specifically, the GFAP isoform, GFAPδ, may form heteromeric intermediate filaments with the isoform GFAPα and vimentin or interact with presenilins [92]. Cysteine residues (Cy294) of GFAP are considered the sensors of redox conditions, mainly affected by oxidative stress leading to impairment of cytoskeletal organization in astrocytes [96]. It should be noted that the GFAPδ-immunopositive cells always express a high level of GFAPα, which is the main isoform of GFAP, but the activity of GFAPδ is critical for the regulation of intermediate filament dynamics [92].

GFAP is phosphorylated by PKA and CaMKII and dephosphorylated by protein phosphatase PP1; therefore, cyclic phosphorylation/dephosphorylation affects the GFAP polymerization in cells [97]. GFAP-immunopositive astrocytes do not cover the whole population of astroglia, i.e., less than half of hippocampal astrocytes are GFAP+ cells even after stimulation of gliosis [98]. Despite this, experimentalists believe that GFAP is a more suitable marker of astrocytes in corpus callosum, cerebral peduncle, and hippocampus if compared to labeling with s100β, which is a Ca^2+^-binding cytosolic and nuclear glial protein found predominantly in the thalamus, and with an N-Myc downstream-regulated gene 2 (NDRG2) protein, which is specific for mature, nonreactive, and nonproliferating astrocytes, particularly in the cortex of mice [11].

Transcriptional control of GFAP expression was extensively studied [99,100]. Astrocytes use a specific mechanism for GFAP gene expression, while neurons utilize different mechanisms to suppress the expression of the GFAP gene [99,100]. STAT3 governs the developmental regulation of GFAP expression in the prenatal brain in cooperation with Smad1. In contrast, in mature astrocytes, GFAP expression is under the control of numerous transcription factors, including AP-1, NF-kB, NF1, Pax3, etc. [100,101]. It was demonstrated that neuron–glia interactions affect this machinery in astroglial cells via glutamate or cytokines acting at astrocytes [102,103].

Epigenetic control of GFAP expression was demonstrated in astrocytes with modified histone acetylation pattern: inhibition of histone deacetylases reduced GFAP expression, elevated the GFAPδ/GFAPα ratio, and increased aggregation of intermediate filaments in primary human astrocytes in vitro [104]. The latter effect was not surprising because GFAPδ expression negatively affects filament formation [105]. In neurogenesis, the GFAP promoter is silenced by DNA methyltransferase I, but when gliogenesis starts, Notch signaling allows GFAP transcription and astrogenesis via inhibition of DNA methyltransferase or directly via activation of the GFAP protomer [106]. Suppression of Notch signaling results in activation of neurogenic potential of astroglial cells, e.g., striatal astrocytes become transcriptionally similar to subventricular zone stem cells and demonstrate no signs of reactive phenotype (including elevated GFAP expression). Then, low expression of GFAP proceeds throughout the neurogenic process [107]. Another mechanism underlying the ability of mature adult astrocytes dedifferentiate to a radial glial cells phenotype, which provides a scaffold for migrating transplanted embryonic neurons. The corresponding changes in RC2 and GFAP expression [108] remain to be evaluated.

***GFAP expression in neurogenic and gliogenic cells.*** GFAP-expressing radial glial cells (RGCs) are descendants of neuroepithelial cells. They appear as cells with bipolar morphology in the mouse embryonic brain at E9 and E10 (at the start point of neurogenesis), presumably allowing appropriate migration of newborn neurons and acting as actual neural stem cells (NSCs). In the adult brain, RGCs can be found in particular regions such as lateral ventricles, hypothalamus, and cerebellum [109]. In mice, GFAP mRNA levels are shown to increase twofold from E15 to E17 and tenfold between E17 and the day of birth; however, the GFAP-expressing cells cannot form neurospheres at E12.5, and only some of the neurosphere-forming cells express GFAP at E15.5 [110]. In humans, at gestational weeks 9–12, expression of GFAP is initiated in RGCs [92], then proliferating neurogenic RGCs (but not resting astrocytes) express the particular splice variant GFAPδ during human brain prenatal development and throughout the postnatal period in the subventricular zone (SVZ) and the hippocampus [111,112]. In mice, the GFAPδ expression increases from E18 to P5 and then decreases to plateau at P25. In contrast to the human brain, all mouse astrocytes in developing and adolescent mouse brains express GFAPδ regardless of their neurogenic potential [113]. In the developing mouse brain, loss of GFAPδ induced by prenatal ethanol exposure provokes the transformation of RGCs into mature astrocytes [114]. Human glial restricted precursor cells (transition state between NSCs and differentiated glial cells) express GFAP in 50% of cells and may increase the expression up to 95% being exposed to the stimuli (i.e., bone morphogenetic protein 4, BMP4) [115].

***GFAP expression in Alzheimer’s disease.*** GFAPδ+ cells resembling RGCs were found in neurogenic zones in patients with Alzheimer’s disease, but in contrast to control cases, there were fewer close contacts between RGCs and blood vessels [112]. The latter suggests impairment of neurovascular unit integrity, aberrant neurogenesis, and pathologically enhanced angiogenesis [116,117].

Progression of Alzheimer’s disease is associated with astroglial atrophy and cytoskeletal rearrangements that are confirmed morphologically and by analyzing the patterns of gene expression: significant decrease of mRNAs encoding for cytoskeletal proteins of myosin, kinesin families, actin, and integrins were detected in the advanced stage of Alzheimer’s disease [118]. If GFAP expression is downregulated, then one could expect changes in the transcriptional patterns of astroglial cells, suggesting more pro-inflammatory phenotype, as was demonstrated in mice lacking GFAP with the transgenic model of Alzheimer’s disease [119].

Astrogliosis due to neuroinflammation is coupled with hypertrophy of astroglial cells and overexpression of GFAP [120], thus, high expression of GFAP (both GFAPα and GFAPδ) and presenilin (PS) (catalytic subunit of γ-secretase) was observed in astrocytes surrounding amyloid plagues in Alzheimer’s disease [121,122]. Because GFAPδ specifically interacts with PS [121], and astroglial cells are equipped with PS [123], enhanced production of amyloid-β should be associated with GFAP overexpression in astroglial cells in Alzheimer’s disease. Progression of local inflammation also contributes to this phenomenon, and it is not a specific feature of Alzheimer’s brain. For instance, excessive gliogenesis due to gliocentric shift in the neurogenesis program in down syndrome results in elevated expression of GFAP that is associated with the action of RAGE receptor ligands, s100β and amyloid precursor protein (APP) [124]. This suggests that neuroinflammation triggers the expression of GFAP. Elevated expression of GFAP in the brain of autistic patients correlates with overexpression of pro-inflammatory cytokines [125]. Similar mechanisms could be in action in the Alzheimer’s brain because high expression of RAGE receptors on neurovascular unit cells [126] or elevated levels of microglia- or astroglia-derived IL-1, IL-6, and TNFα [127] are the characteristics of Alzheimer’s type neurodegeneration.

In summary, utilizing GFAP and a regulatory cell type-specific promoter for efficient transgene expression in astrocytes provides numerous opportunities to target heterogeneous subpopulations of astroglia in the healthy brain and in pathological conditions such as neurodegeneration and neuroinflammation.

### 3.4. Efficient and Restricted AAV-Mediated Transgene Expression in Astrocytes: Does the Promoter Really Matter?

AAV vectors carrying different versions of the GFAP promoter in a cassette (Figure 1) were extensively used in the studies to target astrocytes in rodent and primate brains [45,66,128,129,130,131,132,133]. The conventional 2.2 kb human GFAP promoter (−2163 to +47 from transcription start site), also known as gfa2, was found to drive transgene expression in astrocytes throughout the CNS [66,130]. Different combinations of putative enhancers of *GFAP* gene, named A (−1757 bp), B (−1627 bp), C (−1488 bp), and D (−132 bp), were used for the construction of various GFAP promoter isoforms that displayed greater activity and astrocytic specificity compared to parental gfa2 promoter (Figure 1). Insertion of three additional copies of B enhancer region into gfa2 produced the 2.6 kb gfa2(B)_3_ promoter with improved characteristics but limited application because of its bigger size (Figure 1) [134]. The limited packaging capacity of AAV vectors boosted the generation of truncated versions of the GFAP promoter for applications where size matters. Truncated 1.74 kb GFAP promoter, designed by Meng and colleagues based on gfa2, spanned the enhancer regions A–D and showed robust and highly selective gene expression in murine astrocytes in vivo [131]. In experiments on transgenic mice, the elements in the GFAP promoter that are required for enhanced and targeted gene expression in the brain were studied in detail [99,134]. Excision of the C segment by the juxtaposition of distal (A, B) and proximal (D) enhancer regions yielded the shortest 448 bp GFAP promoter, gfa28 (Figure 1), with very restricted expression in the mouse brain and with broad transduction potential for both astrocytes and neurons [135]. These data brought attention to the region corresponding to enhancer C. It was found that very short sequences of 55 and 45 bp at the beginning of enhancer C may determine region- and cell-specific transgene expression, respectively [99]. Insertion of the initial fragment of enhancer C (C_1_, −1488 to −1256 bp, Figure 1) into gfa28 promoter restored its characteristics to the level that gfa2 has. Specifically, this novel 681 bp GFAP promoter, named gfaABC_1_D (also known as gfa104, Figure 1), displayed greater activity but comparable expression patterns as full-length gfa2 in transgenic animals. Recent studies demonstrated promising potential of gfaABC_1_D promoter for efficient gene expression in astrocytes in vivo using an AAV-based delivery strategy [45,128,129,136,137,138,139].

Some experimental data may create the false impression that swapping a ubiquitous promoter in the cassette to the GFAP-based promoter versions is sufficient to restrict the expression of any produced AAV virus to astrocytic populations [60,66,130,140,141]. However, particular promoter/serotype combinations appear to be more favorable than others to target astrocytes with high efficiency and specificity. For example, the gfaABC_1_D/AAV5 combination was extensively used to trigger robust and efficient astrocytic expression in different brain regions (nucleus accumbens, cortex, hippocampus) and spinal cord [45,128,129,133,139]. Our pilot studies are in agreement with previously published data [45] suggesting that hippocampal astrocytes of rats (not shown) and mice (Figure 2) can be efficiently transduced with the gfaABC_1_D/AAV5 combination (based on the morphology). Successful results were also observed in the brain regions (striatum, cortex, hippocampus) and the spinal cord when gfaABC_1_D-driven constructs were packaged into AAV1, AAV8, or AAV9 [45,136,137,138]. Several studies also suggest that AAV8, 9, and rh43 can be used to mediate astrocytic expression of different constructs, driven by human (2.2 kb) or marmoset GFAP promoter isoforms (1.6 kb, 1.4 kb, 0.6 kb, 0.3 kb, and 0.2 kb), in mouse and primate brains [45,66,132].

In contrast, the human GFAP promoter (hGFAP) did not provide glial-specific expression of a genetic construct in tested brain regions when used with AAV2 viruses [64]. Specifically, in the rat striatum, less than 5% of the total amount of transduced cells were astrocytes, and in the rat hippocampus, the expression was observed almost exclusively in neurons. Combined with either AAV6 or AAV9 serotype, the GFAP-based promoters exhibited weak specificity due to off-target neuronal transduction in rat striatum and substantia nigra, respectively [142,143]. It is also noted that favorable promoter/serotype combinations may differ among brain regions. With regard to astrocytic expression, the gfaABC_1_D/AAV5 works well in some areas (see above) but may be suboptimal in the striatum [143]. At the same time, the gfaABC_1_D/AAV9 seems suitable for astrocyte-specific transgene expression in the striatum [138]. This may imply cellular and molecular heterogeneity of astrocytes among brain regions and their diverse susceptibility to different promoter/serotype combinations. It means that, in some cases, the additional pilot screening of diverse promoter/serotype combinations is required to adjust the experimental protocol for chosen brain area.

Keeping in mind the very limited expression pattern of GFAP throughout the brain, the findings of alternative astrocyte-specific promoters with improved characteristics are of particular interest. Recently, Preston and colleagues reviewed different markers that are commonly used in studies to label astrocytic populations to some degree [144]. Characterization of their promoters may create a basis for efficient genetic targeting of different astrocytic populations; however, due to high cellular and molecular heterogeneity of astrocytes, it seems that it is unlikely that a universal promoter will be found. Promoter specificity of a few potent astrocytic markers (Aldh1l1, hGfap, Glast, Cx30, and Fgfr3) was tested in vivo on transgenic CreERT2 lines [145]. Among others, promising results were shown for transgenic lines with CreERT2 controlled by the native full-length Aldh1l1 promoter [145,146]. However, generally, large cell-specific promoters comprising different regulatory regions require optimization for experiments where size matters. On the basis of the *ALDH1L1* gene from rat and human genomes, two independent groups developed and characterized putative, relatively compact ALDH1L1 promoters (Figure 3) suitable for the AAV expression cassette [142,147]. Surprisingly, ALDH1L1-driven constructs packaged into AAVs with known astrocytic tropism were strongly expressed in neurons and some astrocytes in a region-dependent manner, ranging from mostly astrocytic expression in the thalamus to mostly neuronal expression in mPFC, hippocampus, and substantia nigra [142,147].

Extensive characterization of promoter organization was also provided for other astrocytic markers, such as Glast (encoded by *Slc1a3* gene) and Cx30 (encoded by *GJB6* gene), in different cell lines, although their specificities were not tested in glial cells in combination with AAVs [148,149]. Detailed analysis of the *GJB6* promoter region with corresponding regulatory elements, encompassing −461 to +33 bp from TSS, was performed by Essenfelder and colleagues (Figure 3) [149]. In another work, serial deletions of upstream region −3830 to +450 bp of mouse *Slc1a3* gene revealed the presence of a highly effective compact promoter between −636 to −169 bp (Figure 3) [148]. The authors also found some regulatory motifs located in the *Slc1a3* promoter (−200 to −195 bp; −111 to −106 bp, Figure 3) that are characteristic of housekeeping genes, which may be relevant for stabile widespread expression of *Slc1a3*-driven genetic constructs. Interestingly, the *Slc1a3* promoter region −3830 to −636 bp contains negative regulatory elements that may explain recently observed low gene expression driven by the native full-length *Slc1a3* (Glast) promoter in several brain areas of transgenic mice [145]. Altogether, these findings show the potential of truncated versions of Glast and Cx30 promoters for AAV-based targeting of glial cells. However, additional verification of their specificity in vitro and in vivo is strongly recommended.

To summarize, a proper combination of particular AAV serotypes naturally predisposed to transduce astrocytes with astrocyte-specific promoters can reduce off-targeting neuronal populations.

## 4. Conclusions

This review aimed to analyze the principles and the perspectives of using the genetic constructs based on AAV vectors for the regulation of astrocytes activity. The extreme heterogeneity of astrocytes, consisting of local populations with different morphological, functional, and molecular characteristics, complicates the rational design of an ideal viral vector for specific aims. As a consequence, it results in uneven transduction efficacy across brain regions and between the species.

AAVs of different serotypes have no unique selectivity toward astrocytes; however, it appears that AAV5, AAV8, and AAV9 are naturally predisposed to target astrocytes better than others. Application of AAVs, specifically AAV8 and AAV9, provides variable results depending on experimental conditions, AAV production protocols, and region- and cell-specific characteristics in the injection area. Application of natural AAV serotypes for cell-specific transduction has obvious limitations that can be bypassed by generating synthetic “designer” AAV variants with specified, enhanced properties through rational engineering of AAV capsids or using a directed evolution approach [67,73,150,151,152]. Realizing these strategies for developing novel synthetic serotypes for astrocytic targeting has excellent potential for basic and translational neuroscience. Recent advances in high-throughput screening and computational modeling approaches may provide a novel insight into proper validation of generated synthetic astrocyte-specific AAV serotypes [153,154].

Another critical determinant of efficient and specific targeting of genetic constructs refers to promoter properties. The utilization of AAV vectors driven by ubiquitous promoters may not provide the expected transduction pattern. In contrast, the use of cell-specific promoters can undoubtedly increase the selectivity of the transgene expression. However, large cell-specific promoters composed of different regulatory regions require optimization for experiments where the size of the genetic construct matters. As a result, such optimization can compromise their specificity [142,147]. Elimination of the observed off-target neuronal expression can be achieved by combining glial-specific promoters in the AAV cassette with particular microRNAs that provide selective gene silencing in neuronal populations [143]. However, this significantly reduces the packaging capacity of AAV vectors. The advanced strategy of substituting large promoters with unique enhancer sequences to drive transgene expression selectively in neurons in a restricted brain area was tested recently and showed great potential in combination with AAV vector as well [155,156,157]. Similar approaches employed for description and characterization of specific astrocytic enhancers have great potential for developing methods for highly selective regulation of astrocytes in specific regions or across the brain.

Swapping of a ubiquitous promoter in the AAV cassette to the glial promoter is insufficient to restrict the expression of any produced AAV virus to astrocytic populations itself [60,66,130,140,141]. Some glial promoters provided relatively strong neuronal transduction in the injected brain areas in particular conditions. Certain promoter/serotype combinations seem to be more favorable than the others to target astrocytes with high efficiency and specificity. Based on published data and our pilot data, the gfaABC1D/AAV5 combination has great potential in targeting astrocytes. Combined with glial promoters, AAV1, AAV8, and AAV9 also demonstrated sufficient astrocytic transduction. Therefore, the proper combination of AAV serotypes that are naturally predisposed to transduce astrocytes with astrocyte-specific promoters can reduce the undesired targeting of neuronal populations.

In summary, using genetic constructs for efficient stimulation of astrocytes with the aim to activate neurons, one must take into account: (i) the type of target glial cells including regional and functional specificity; (ii) the serotype/pseudotype of the recombinant AAV; (iii) the specificity of a promoter; (iv) the combinations of serotype and promoter; (v) the dependence of released gliotransmitter on the parameters of stimulation. Combining these parameters makes the task challenging, but solving these questions would give us new ways to modulate astroglial activity in a healthy brain or compensate for pathological changes in the brain.

## Figures and Tables

**Figure 1 cells-10-01600-f001:**
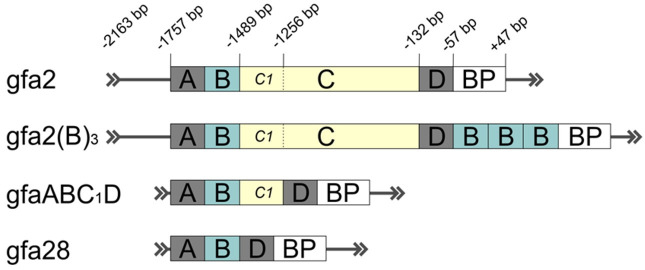
Schematic representation of characterized promoter isoforms for astrocytic targeting constructed based on human *GFAP* gene (for details, see [99,134]). Letters A–D indicate enhancer regions, BP–basal promoter. Relative distances from the transcriptional start site (TSS) are provided in bp (base pairs). Arrows show the direction of transcription.

**Figure 2 cells-10-01600-f002:**
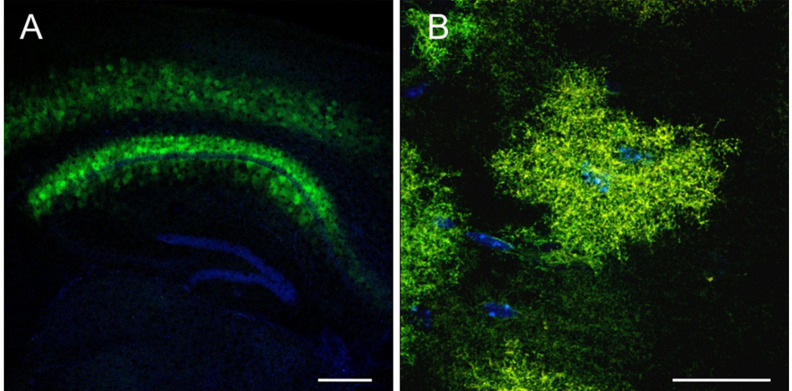
High-resolution confocal images of mouse hippocampal astrocytes at low ((**A**), scale bar: 500 µm) and high magnification ((**B**), scale bar: 30 µm), transduced by AAV5 viruses with a reporter gene (opto-a1AR, green) driven from the astrocytic gfaABC_1_D promoter. Nuclei are labeled with DAPI (blue).

**Figure 3 cells-10-01600-f003:**
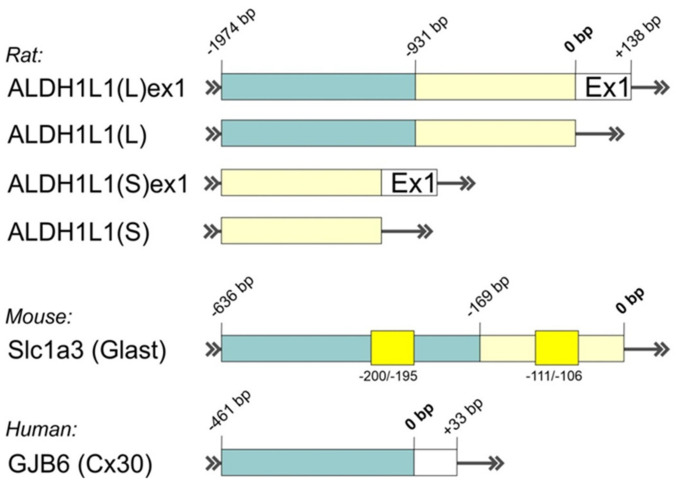
Schematic representation of putative promoter isoforms for astrocytic targeting constructed based on rat *Aldh1l1* gene (for details, see [142]); mouse *Slc1a3* gene (for details, see [148]), and human *GJB6* gene (for details, see [[149]). Relative distances from the transcriptional start site (TSS) are provided in bp (base pairs). Arrows show the direction of transcription. Yellow boxes represent important regulatory regions. Ex1-the initial part of the first exon.

**Table 1 cells-10-01600-t001:** A list of various promoters, genes of interest (opsins), fluorescent reporter proteins, and regulatory elements that can be used to design the genetic construct for efficient transgene expression. For each element, relative length (kb), source, and/or Addgene catalog indexes (#) are provided. GOI—gene of interest; PRE—post-transcriptional regulatory element; polyA—polyadenylation signal.

	Name of Element	Length, kb	Source	Addgene #
**Promoter**	**Ubiquitous promoters**			
	CBA (CAG)	1.6	Synthetic	
	CBA mini	0.8	Synthetic	
	CBh	0.8	Synthetic	
	CMV	0.8	Viral	
	**Validated astrocytic promoters**			
	gfa2(B)3	2.6	Human	#53132
	gfa2 (hGFAP)	2.2	Human	#53126
	gfaABC1D (gfap104)	0.7	Human	#53131, #122630
	gfa28	0.45	Human	#53130
	**Putative astrocytic promoters**			
	AldH1L1	0.9–2.1	Human, Rat	
	Slc1a3 (Glast)	0.64	Mouse	
	GJB6 (Cx30)	0.5	Human	
**GOI (Opsin)**	Melanopsin	1.4	Human	#122630
	Opto-a1AR	1.3	Chimeric	#20947
	ChR2(H134R)	0.94	Algae *	#112496
**Fluorescent protein**	tdTomato	1.4	Anthozoa *	
	EGFP	0.72	Hydrozoa *	
	EYFP	0.72	Hydrozoa *	
	mCherry	0.71	Anthozoa *	
**PRE**	WPRE	0.6	Viral	
	WPRE3	0.25	Viral *	#61463
**polyA**	bGH	0.22	Bovine	
	SV40 late	0.13	Viral	
	hGH	0.48	Human	

* Elements with some modifications (mutations etc.).

## Data Availability

Original data are available on request.

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
