# Peer review of "Genetic Constructs for the Control of Astrocytes’ Activity"

_cells, 2021, doi:10.3390/cells10071600_

Round 1

Reviewer 1 Report

The proposed review is devoted to the evaluation of the prospects of using genetic constructs based on AAV vectors to regulate astrocyte activity, including the optogenetic methods.
Undoubtedly, the article’s topic is highly relevant and is at the forefront of research in neurobiology. As active participants in the synaptic transmission and the functioning of neuron-glial networks, astrocytes are attracting more and more researchers’ attention.
The review covers in detail the available data on the creation of genetically engineered construct designed for astrocyte transduction. Of particular interest is the information on the features of various astrocyte subtypes that determine the methodological approaches to creating vectors. In addition, a comparative assessment of different AAV serotypes and different promoters for targeting the vector to specific astrocyte populations in different parts of the brain is presented. All this makes the article interesting for the Cells’ readers.
In section "2.2. Measuring astroglial activity", the authors mention that “some recent data suggest that calcium dependence in the release of gliotransmitters in astrocytes associated with specific neural circuits might be overestimated,” but they do not provide any details of this problem. Since most of the studies of astrocytic activity are related specifically to the study of calcium dynamics, a more detailed description of the issue mentioned by the authors would be interesting and enrich the article.

Author Response

Reviewer # 1: In section "2.2. Measuring astroglial activity", the authors mention that “some recent data suggest that calcium dependence in the release of gliotransmitters in astrocytes associated with specific neural circuits might be overestimated,” but they do not provide any details of this problem. Since most of the studies of astrocytic activity are related specifically to the study of calcium dynamics, a more detailed description of the issue mentioned by the authors would be interesting and enrich the article.

Authors: We thank the Reviewer for this comment, and we added some information on the role of Ca2+-dependent exocytosis of glutamate from astroglial cells.

We are very much grateful for all the suggestions and comments provided by the Reviewer that help us to improve the manuscript.

Reviewer 2 Report

Dear Editor,

The manuscript by Borodinova et al. discusses principles and perspectives of using the genetic constructs based on AAV vectors to regulate astrocytes’ activity.

The review is comprehensive, informative and up-to-date (in most parts). Authors were successful in providing some well compiled opinions and summaries. The mechanistic figures and table can be a good starting point for future studies and will be of interest for Cells readers and beyond.

However, there is a number of major and minor points that would need to be addressed in order to improve the quality of this paper before it can be accepted for publication.

General:

- This review overlooked some essential and up-to-date work regarding the role of astrocytes. I have made some suggestions below but authors are encouraged to consider citing updated references throughout the review, whenever possible.

Major:

-Authors need to mention the work by Qian et al. Nature 2020 where they have beautifully shown that the conversion of midbrain astrocytes to dopaminergic neurons, which provide axons to reconstruct the nigrostriatal circuit. This can be added in line 222 where the authors mention neurodegenerative diseases so they can expand it to include Parkinson’s. Reference:

https://pubmed.ncbi.nlm.nih.gov/32581380/

- The author does not reference a key study from 2020, demonstrating that the development of edema following injury-induced hypoxia is aquaporin-4 dependent. This breakthrough study showed that essential role of targeting astrocytes in CNS injures. Reference to ne included: https://www.cell.com/cell/fulltext/S0092-8674(20)30330-5.

This role has been recently been confirmed by the work of Sylvain et al BBA 2021 which has demonstrated that targeting AQP4 effectively reduces cerebral edema during the early acute phase in in ischemic stroke. They have also shown a link to brain energy metabolism as indicated by the increase of glycogen levels. Reference to be included:

https://pubmed.ncbi.nlm.nih.gov/33561476/

- Astrocytes are known to have polarised pattern of expression for GFAP and AQP4. However, most of the results in current literature are either from 2D models cell culture or animal studies. Authors need to expand the discussion of this point where they mentioned “Human and higher-order primate protoplasmic astrocytes are larger and express more GFAP than rodent protoplasmic astrocytes [16]”. This could include, but not limit to, the use of humanized self-organized models, organoids, 3D cultures and human organ-on-a-chip platforms especially those which are amenable for advanced imaging such as TEM and expansion microscopy since they enable real-time monitoring of astrocytic cellular mechanisms. This is quite important since AQP4 in astrocytes, for example, has polarised expression which has proven to be varied between 2D and 3D cultures. References to be included:

https://pubmed.ncbi.nlm.nih.gov/30165870/

https://pubmed.ncbi.nlm.nih.gov/33117784/

https://pubmed.ncbi.nlm.nih.gov/31889243/

-2.2 number ii “ii) metabolic changes in astrocytes (lactate production, K+ uptake, mitochondrial activity, etc.) caused by their activation [19]”. This is rather a simplistic view. Authors need to mention the astrocyte‐neuron lactate shuttle (ANLS) hypothesis postulated in 1994 (Pellerin and Magistretti 1994). According to this, astrocytes serve as a ‘lactate source’ whereas neurons serve as a ‘lactate sink’. Moreover, the opposition by Bak and colleagues who argued that oxidative metabolism of lactate within neurons only occurs during repolarization (and in the period between depolarizations) rather than during neurotransmission activity. The emerging role of astrocytes has helped in settling this debate in favour for ANLS hypothesis. References to be included:

https://pubmed.ncbi.nlm.nih.gov/31318452/

https://pubmed.ncbi.nlm.nih.gov/19393013/

https://pubmed.ncbi.nlm.nih.gov/7938003/

Minor:

-End of discussion and towards the conclusion: Authors needs to point out to the recent advances in applying the use of high-throughput screening and computer-aided drug design as have been nicely reviewed by Aldewachi et al 2021 and Salman et al 2021 as they can provide a novel insight that can support specific target validation for astrocyte-specific AAVs in future studies. References to be included:

https://pubmed.ncbi.nlm.nih.gov/33672148/

https://pubmed.ncbi.nlm.nih.gov/33925236/

Best.

Author Response

Major:

Reviewer # 2: Authors need to mention the work by Qian et al. Nature 2020 where they have beautifully shown that the conversion of midbrain astrocytes to dopaminergic neurons, which provide axons to reconstruct the nigrostriatal circuit. This can be added in line 222 where the authors mention neurodegenerative diseases so they can expand it to include Parkinson’s. Reference:

https://pubmed.ncbi.nlm.nih.gov/32581380/

Authors: We thank the Reviewer for this and other comments, and we added the suggested reference into the text.

Reviewer # 2:  The author does not reference a key study from 2020, demonstrating that the development of edema following injury-induced hypoxia is aquaporin-4 dependent. This breakthrough study showed that essential role of targeting astrocytes in CNS injures. Reference to ne included: https://www.cell.com/cell/fulltext/S0092-8674(20)30330-5.

Authors: we agree with the Reviewer, and we added the suggested reference into the text.

Reviewer # 2: This role has been recently been confirmed by the work of Sylvain et al BBA 2021 which has demonstrated that targeting AQP4 effectively reduces cerebral edema during the early acute phase in in ischemic stroke. They have also shown a link to brain energy metabolism as indicated by the increase of glycogen levels. Reference to be included:

https://pubmed.ncbi.nlm.nih.gov/33561476/

Authors: we agree with the Reviewer, and we added the text and the suggested reference into the text.

Reviewer # 2: Astrocytes are known to have polarised pattern of expression for GFAP and AQP4. However, most of the results in current literature are either from 2D models cell culture or animal studies. Authors need to expand the discussion of this point where they mentioned “Human and higher-order primate protoplasmic astrocytes are larger and express more GFAP than rodent protoplasmic astrocytes [16]”. This could include, but not limit to, the use of humanized self-organized models, organoids, 3D cultures and human organ-on-a-chip platforms especially those which are amenable for advanced imaging such as TEM and expansion microscopy since they enable real-time monitoring of astrocytic cellular mechanisms. This is quite important since AQP4 in astrocytes, for example, has polarised expression which has proven to be varied between 2D and 3D cultures. References to be included:

https://pubmed.ncbi.nlm.nih.gov/30165870/

https://pubmed.ncbi.nlm.nih.gov/33117784/

https://pubmed.ncbi.nlm.nih.gov/31889243/

Authors: we agree with the Reviewer, and we added the text and the suggested references into the text.

Reviewer #2:  2.2 number ii “ii) metabolic changes in astrocytes (lactate production, K+ uptake, mitochondrial activity, etc.) caused by their activation [19]”. This is rather a simplistic view. Authors need to mention the astrocyte‐neuron lactate shuttle (ANLS) hypothesis postulated in 1994 (Pellerin and Magistretti 1994). According to this, astrocytes serve as a ‘lactate source’ whereas neurons serve as a ‘lactate sink’. Moreover, the opposition by Bak and colleagues who argued that oxidative metabolism of lactate within neurons only occurs during repolarization (and in the period between depolarizations) rather than during neurotransmission activity. The emerging role of astrocytes has helped in settling this debate in favour for ANLS hypothesis. References to be included:

https://pubmed.ncbi.nlm.nih.gov/31318452/

https://pubmed.ncbi.nlm.nih.gov/19393013/

https://pubmed.ncbi.nlm.nih.gov/7938003/

Authors: we agree with the Reviewer, and we added the text and the suggested references into the text.

Minor:

-End of discussion and towards the conclusion: Authors needs to point out to the recent advances in applying the use of high-throughput screening and computer-aided drug design as have been nicely reviewed by Aldewachi et al 2021 and Salman et al 2021 as they can provide a novel insight that can support specific target validation for astrocyte-specific AAVs in future studies. References to be included:

https://pubmed.ncbi.nlm.nih.gov/33672148/

https://pubmed.ncbi.nlm.nih.gov/33925236/

 Authors: thank you for suggestion. We added required references into the “conclusion” section. We are very much grateful for all the suggestions and comments provided by the Reviewer that help us to improve the manuscript.

Round 2

Reviewer 2 Report

Dear Editor,

The authors have successfully addressed the majority of my comments and concerns in order to improve the quality of the manuscript.

I believe that the corrections, additional sections and updated references, have contributed to enhancing the clarity of the manuscript, which I can now endorse for publication.

All the best!